# Predicting the Sensory Consequences of Self-Generated Actions: Pre-Supplementary Motor Area as Supra-Modal Hub in the Sense of Agency Experience

**DOI:** 10.3390/brainsci10110825

**Published:** 2020-11-07

**Authors:** Silvia Seghezzi, Laura Zapparoli

**Affiliations:** 1Psychology Department and NeuroMI—Milan Center for Neuroscience, University of Milano-Bicocca, 20126 Milan, Italy; 2PhD Program in Neuroscience, School of Medicine and Surgery, University of Milano-Bicocca, 20126 Milan, Italy; 3fMRI Unit, IRCCS Istituto Ortopedico Galeazzi, 20161, Milan, Italy

**Keywords:** motor awareness, sense of agency, intentional binding, fMRI, pre-SMA

## Abstract

Sense of agency refers to the feeling that one’s self-generated action caused an external environment event. In a previous study, we suggested that the supplementary motor area (SMA), in its anterior portion (pre-SMA), is a key structure for attributing the sense of agency for the visual consequences of self-generated movements. However, real-life actions can lead to outcomes in different sensory modalities, raising the question of whether SMA represents a supra-modal hub for the sense of agency. Here, we compared the agency experience for visual and auditory outcomes by taking advantage of the intentional binding effect (IB). We observed discrete time-windows for the agency manifestation across different sensory modalities: While there was an IB at 200 ms delay between the action and the visual outcome, a time compression was observed when the auditory outcome followed the action by 400 ms. The magnitude of the IB was mirrored by meaningful brain activity in the pre-SMA but only at the specific delay when a sizeable IB was seen. We conclude that attributing consequences of self-generated movements to one’s action is based on similar mechanisms across sensory modalities and that those mechanisms are related to the functioning of the motor system.

## 1. Introduction

According to theories on sensorimotor control [1,2,3], predicting actions’ sensory consequences is essential to accomplish the current motor programs and efficiently process the incoming sensory stimuli. It also contributes to attributing the external effects of self-generated movements to oneself (“I did that”). This latter ability has been defined as “sense of agency, and it arises when predictions about the possible consequences of a voluntary action match the actual external outcome [4].

In a recent study, we explored the brain mechanisms underlying the sense of agency through an fMRI experiment based on visual stimuli [5]. We used the intentional binding phenomenon [6], an implicit measure of the agency experience whereby the temporal interval between voluntary actions and their effects is perceived to be shorter than its real lasting. We showed that the sense of agency is associated with a specific brain network, including the pre-supplementary motor area (pre-SMA) and dorsal parietal cortex, which activity was positively correlated with the degree of the intentional binding effect. We also found that repetitive transcranial magnetic stimulation (rTMS) over the pre-SMA affected the sense of agency by extending the normal time-window for binding manifestation. This only happened when the stimulation was applied before the action execution when predictions about action consequences are generally made [4]. In this condition, there was a sizeable intentional binding effect in the case of a stringent 200 ms temporal contiguity between action and outcome and when the outcome followed the action by 400 ms [5]. These findings suggest the crucial (and causal) role of the pre-SMA in the predictive mechanisms underlying the sense of agency.

There is independent evidence of a general link between the SMA, with particular reference to its anterior portion, and the sense of agency experience. For example, EEG evidence showed that self-initiated movements following early readiness potentials—which has been related to the activity of the pre-SMA [7]—result in a stronger binding effect compared to positive potentials [8]. Non-invasive brain stimulation evidence suggested that transcranial direct current stimulation over the pre-SMA reduces the intentional binding effect towards auditory outcomes [9]. In another study, continuous theta-burst stimulation over pre-SMA was shown to reduce the temporal linkage between a voluntary key-press action and a subsequent electrocutaneous stimulus [10]. Last, a study with patients with corticobasal syndrome showed that functional connectivity patterns between the pre-SMA and the prefrontal cortex in resting conditions change according to the intentional binding effect [11]. This evidence supports the view that the brain activity associated with the motor plan’s generation is somehow crucial for giving rise to a sense of agency [1,2,3]. Notably, the relationship between agency and the SMA/pre-SMA activity seems to remain true irrespective of the specific features of the outcomes: It has been shown for visual [5], auditory [8,12], and tactile events [10], giving rise to the possibility of a supra-modal nature of this mechanism.

It is certainly possible that SMA, precisely in its anterior portion, represents a unique supra-modal predictive mechanism for the agency attribution. Pre-SMA is a key structure for preparing and initiating voluntary actions, showing greater activity for self-initiated compared with externally-triggered actions [13,14]. It is also strongly connected with prefrontal cortices [15,16], and it has a specific role in the performance of complex motor tasks, such as the alternation of motor plans, acquisition of new motor skills, and motor selection [17,18,19]. For these reasons, pre-SMA may contribute to a pre-motoric, supra-modal generation of the sense of agency.

However, previous research has mainly focused on the agency experience in one sensory modality only. None of the aforementioned neuroimaging studies directly tested the pre-SMA role on the agency experience across different modalities through the same paradigm applied to the same sample of participants.

This study further explores the link between agency and SMA activity in a new behavioral and fMRI experiment. In particular, we assessed the SMA role in the sense of agency feeling for action consequences in a different sensory modality compared to that tested in our previous study [5]. Specifically, while we used visual feedback in the previous study [5], the sensory effect produced by the voluntary action in this task was an auditory tone. We used an identical experimental paradigm based on a temporal judgment task administered to the same sample of participants in an fMRI setting. At a behavioral level, we looked for possible differences between the visual and auditory domains in the behavioral manifestation of the sense of agency experience, i.e., the intentional binding effect at different action-outcome delays. At a neurofunctional level, we tested the hypothesis that the activity of SMA covaried with the individually measured intentional binding effect in specific time-windows between actions and outcomes.

We hypothesized that SMA, particularly in its anterior portion, is a supra-modal area in the sense of agency attribution. Accordingly, as for the visual consequences of actions, we would expect to see a linear relationship between the pre-SMA activity and the magnitude of the intentional binding effect also for auditory consequences. This would generalize the relationship between pre-SMA and sense of agency by excluding the possibility that previous results could be somehow dependent on the used paradigm’s specific features. Alternatively, we would expect no correlation between the pre-SMA and the sense of agency measure. In this case, different areas could be correlated to the temporal linkage between actions and auditory consequences.

## 2. Materials and Methods

### 2.1. Participants

Twenty-five healthy adult participants (mean age, 25.7 ± 3.8 years; mean education level, 15.6 ± 2.5 years) participated in the experiment. Participants were the same who were included in a previous paper from our group [5].

Data from one participant were excluded from the fMRI analysis due to excessive head movement. The final sample was made of 24 subjects (mean age, 25.4 ± 3.5 years; mean education level, 15.5 ± 2.5 years), all right-handed as evaluated by the Edinburgh handedness inventory [20]. The study protocol was accepted by the local Ethics Committee (IRCCS San Raffaele of Milan; Prot. SOA, 149/INT/2016), and informed consent was acquired from all subjects according to the Helsinki Declaration (1964). All participants participated in the study after the experimental procedure had been fully described.

### 2.2. Procedure

fMRI scans were acquired during the execution of a temporal judgment task.

There were two different tasks: an auditory task and a visual task. Tasks were performed in the same experimental session in separate blocks (one auditory block and one visual block). Blocks were presented in a counterbalanced order across participants to exclude any possible effect of the task order. Each block had an event-related interleaved structure with 60 trials and lasted approximately twelve minutes.

Before the experiment, subjects performed a training session composed of 10 trials for each task, when they received feedback on their accuracy.

Stimuli presentation was controlled by Cogent 2000 MATLAB Toolbox (MathWorks, Natick, MA, USA). Auditory stimuli were delivered through MRI-compatible headphones (Resonance Technology Inc., Northridge, CA, USA). Visual stimuli were presented using VisuaStim fiber-optic goggles (600 × 800 pixel resolution) (Resonance Technology Inc., Northridge, CA, USA). Responses were recorded through response boxes placed under the participant’s hands (Resonance Technology Inc., Northridge, CA, USA).

### 2.3. Experimental Task

Participants performed a temporal judgment task under active and passive conditions (see also [5,21]).

Each trial started with an instruction indicating the nature (active or passive) of the trial: In the visual task, the color of the basis of a visually presented lightbulb (green: active condition; red: passive condition); in the auditory task, an auditory instruction (“go”: active condition, “no-go”: passive condition).

During the active trials, participants were instructed to press a button with their right index finger at their own time after the instruction’s presentation. This was done to elicit a well-prepared, self-initiated button press, rather than an automatic movement as a reflex to the instruction. During the passive trials, participants were instructed to stay still while an experimenter pressed their right index finger to induce a passive movement. In both conditions, the button press caused an action-consequence: the lightening of the visually presented lightbulb in the visual task, a pure tone (1000 Hz) in the auditory task. The consequence was presented after a variable delay of 200, 400 or 600 ms. The feedback lasted 500 ms. After 2000 ms, participants then judged the perceived time interval between their button press and the action-consequence (the lightening on of the lightbulb or the auditory tone). Judgments were reported using a visual analog scale at which participants responded using a five-key response keypad with their left hand. Participants had up to 4 s to give their response. They used their fingers to choose one of five possible options: 1, 200, 400, 600, or 800 ms. The lowest and the highest options were used to allow subjects to both underestimate and overestimate each presented temporal delay. See Figure 1.

Each task included 60 trials: 30 active and 30 passive trials, with ten trials for each action-outcome delay (200/400/600 ms).

The inter-stimulus interval randomly varied between 1500 ms and 2500 ms.

### 2.4. Statistical Analyses of the Behavioral Data

We analyzed the behavioral data collected during the fMRI session using the software Jamovi (The jamovi project, Sydney, Australia). Retrieved from https://www.jamovi.org.

As an implicit measure of the sense of agency, we took advantage of the intentional binding phenomenon [6]. In particular, for each trial, we calculated the “time compression” (TC), i.e., the difference between the estimated and the actual duration of the action-outcome interval: The larger the compression (i.e., more negative values) in the active condition than the passive one, the stronger the sense of agency.

We first carefully inspected our data for the identification of within-subject and between-subject outliers. We excluded from our analyses time compression values that exceeded, in both directions, two standard deviations the mean within-subject or between-subject values. We excluded 4.8% of trials.

We first checked our data distribution using the Cullen and Frey graph [22]. This graph provides the best fit for an unknown distribution according to skewness level and kurtosis. Our data showed a normal distribution.

We then tested for any possible difference in TC between the two different sensory modalities at the different action-outcome delays. TC values represented the dependent variable of the model. The independent variables of the model were the within-subject factors “Condition” (active/passive), “Delay” (200/400/600 ms), and “Modality” (Visual/Auditory). We tested this statistical model by using linear mixed models with random intercept. Planned Bonferroni-corrected post-hoc tests were used to explore significant interactions.

### 2.5. fMRI Data Acquisition and Analysis

MRI scans were performed using a 1.5 T Siemens Avanto scanner (Siemens Healthcare s.r.l., Milano Italia), equipped with gradient-echo echo-planar imaging (flip angle 90°, TE = 40 ms, TR = 2000 ms, FOV = 250 mm, matrix = 64 × 64). The number of the collected fMRI volumes ranged from 210 to 223 volumes on the basis of the individuals’ responses. The first 15 volumes, corresponding to the instructions, were not included in the analyses.

### 2.6. fMRI Data Analysis

Here, we focused on the fMRI data from the auditory task. fMRI data collected during the visual task are published in our recent study [5] and are not included in the analyses. They are reported in the present paper for visualization only.

Given our strong a-priori hypothesis, we focused our analyses on SMA. In particular, we tested for any relation between the time compression measure and the BOLD activity of this area during the auditory task, at different action-outcome delays.

#### 2.6.1. Pre-Processing

After the image reconstruction, raw data conversion from DICOM to the NIFTI format was computed using MRIcron (www.mricro.com). Subsequent data analyses were performed in MATLAB R2014a (Mathworks, Natick, MA, USA) with the software Statistical Parametric Mapping (SPM12, Wellcome Department of Imaging Neuroscience, London, UK). fMRI scans were first realigned to the first image of the run to account for any head movement. Then the structural T1 image was coregistered to the functional mean image for a more precise normalization; the unified segmentation and nonlinear warping approach of SPM12 was used to normalize structural and functional images to the MNI (Montreal Neurological Institute) template to allow group analyses of the data [23,24]. The data matrix was interpolated to generate 2 × 2 × 2 mm voxels. The stereotactically normalized scans were smoothed using a Gaussian filter of 10 × 10 × 10 mm to increase the signal-to-noise ratio, making the data suited for cluster-level correction for multiple comparisons [25].

#### 2.6.2. First Level Fixed-Effect Analyses

The BOLD signal associated with each task condition was treated with a convolution with a canonical hemodynamic response function (HRF) [26]. Global differences in the fMRI signal were removed from all voxels with proportional scaling. High-pass filtering (128 s) was adopted to remove artefactual contributions to the fMRI signal. For each participant, a fixed-effect analysis was performed to characterize the BOLD response associated with each experimental condition before entering the relevant individual contrast images into a random-effect analysis.

At the first level, we characterized the brain activity between the auditory instruction and the subsequent evoked tone.

We included one regressor for each condition (active and passive trials) and each action-outcome delay (200/400/600 ms), for a total of six regressors. Moreover, brain activity occurring between the appearance of the evaluation scale and the response was modeled separately for each delay and condition and treated as non-interest regressors.

Last, the realignment parameters were added as non-interest regressors in order to account for the impact of motion artifacts on the estimates of the beta parameters.

For each participant and each action-outcome delay, we created a contrast image of the comparison Active condition > Passive condition, for a total of three contrast images per participant.

fMRI data collected during the visual task are not included in the analyses, but they are reported in the present paper for visualization only.

#### 2.6.3. Second Level Random Effect Analysis

Since the intentional binding effect, like many other perceptual illusions, is usually observed as a mean [6], while there is a large trial-to-trial variability, we favored an interindividual approach rather than an intraindividual one as in Kuhn et al. [12]. In particular, we performed a second level analysis in which each contrast image (Active condition > Passive condition, one image for each subject for each action-outcome delay 200, 400, and 600 ms, for a total of three contrast images per subject) was entered in a one-way ANCOVA analysis, conforming to a random-effect approach [27], with TC values as a covariate of interest.

We explored the effect of delay-specific TC measure on the delay-specific contrast images to test the hypothesis that the SMA activity covaried with the TC measure in specific time-windows. This also accounts for the single-subject variability since some individuals do not normally show the effect (see, for example, [10]). It is important to note that because the contrast images used in this analysis resulted from the difference between active and passive trials, a differential TC measure between active and passive trials was used as a regressor here.

We used as an explicit mask a region of interest, created using bilateral anatomical masks based on the automated anatomical labeling (AAL) atlas [28] of SMA. The explicit mask was applied to the group brain level.

We then compared the correlation coefficients obtained for each delay by using the Fisher r-to-z transformation.

All the reported results survive a correction for multiple comparisons: We used the nested-taxonomy strategy recommended by Friston et al. [29], including regional effects meeting either a cluster-wise or voxel-wise FWER correction. The voxel-wise threshold applied to the statistical maps before the cluster-wise correction was *p* < 0.001 uncorrected, as recommended by Flandin and Friston [25]. For clusters significant at the *p* < 0.05 FWER-corrected level, we also report the other peaks at *p* < 0.001.

## 3. Results

### 3.1. Behavioral Results

We found a significant effect of the factor “Condition” (F(1,2614) = 18.7; *p* < 0.0001), “Delay” (F(2,2614) = 28.23; *p* < 0.0001) and “Modality” (F(1,2614) = 10.53; *p* = 0.003) and a significant “Condition*Delay*Modality” interaction (F(2,2614) = 3.33; *p* = 0.04). The interactions “Condition*Delay” (F(2,2614) = 2.44; *p* = 0.09), “Condition*Modality” (F(1,2614) = 0.35; *p* = 0.56) and “Delay*Modality” (F(2,2614) = 0.04; *p* = 0.96) were not significant. The Condition * Delay * Modality interaction was explored with planned post-hoc comparisons.

Since the intentional binding effect is based upon specific differences in Time Compression between active and passive conditions [6], whereby the passive condition represents the baseline, we explore differences in TC values between active and passive conditions at different action-outcome delays, separately for each modality.

In the visual task, the perceived TC was significantly higher in the active trials compared with the passive ones when there was a temporal contingency of 200 ms between the movement and the lightening of the lightbulb. For details, see Table 1 and Figure 2a.

In the auditory task, the perceived TC was significantly stronger in the active trials compared with the passive ones at 400 ms of delay between the movement and its consequence, with a trend towards significance at 600 ms. For details, see Table 1 and Figure 2b.

### 3.2. fMRI Results

#### 3.2.1. Effect of the Differential Delay-Specific TC Measure on the Delay-Specific Contrast Images (Active > Passive Trials) at 200 ms Action-Outcome Delay

No cluster showed a significant relationship with the differential TC values for 200 ms of delays between action and the outcome. See Figure 3a.

#### 3.2.2. Effect of the Differential Delay-Specific TC Measure on the Delay-Specific Contrast Images (Active > Passive Trials) at 400 ms Action-Outcome Delay

We identified one cluster in the pre-SMA that showed a significant linear relationship between the differential TC values of individual participants and the BOLD signal during the task (x = 4, y = 20, z = 62, z score = 4.08, FWE-corrected *p* = 0.01, peak-level, FWE-corrected *p* = 0.018, cluster-level). Specifically, more negative TC values (estimated time interval shorter than the real interval, greater intentional binding) in the active than passive conditions were associated with higher BOLD response in these areas. See Figure 3b.

#### 3.2.3. Effect of the Differential Delay-Specific TC Measure on the Delay-Specific Contrast Images (Active > Passive Trials) at 600 ms Action-Outcome Delay

No cluster showed a significant relationship with the differential TC values for 600 ms of delays between action and the outcome. See Figure 3c.

The Fisher r-to-z transformation showed that the regression coefficient (r), indicating the strength of the association between the BOLD activity of the pre-SMA and the individually measured time compression values, was significantly higher for the 400 ms of action-outcome delay than the same coefficients calculated for the 200 and 600 ms of action-outcome delay (200 ms *r* = −0.02, 400 ms *r* = −0.6, 600 ms *r* = −0.004; 200 ms *r* vs. 400 ms *r*: z-score = 2.18, *p*-value = 0.015; 400 ms *r* vs. 600 ms *r*: z-score = 2.23, *p*-value = 0.013).

To further explore whether there was a significant overlap between the regions that showed an association with the behavioral time compression values in the specific tasks, we run a conjunction analysis. The results showed a significant cluster in the pre-SMA (x = 2, y = 20, z = 64, z score = 2.75, FWE-corrected *p* = 0.02, peak-level, FWE-corrected *p* = 0.037, cluster-level, small-volume corrected).

## 4. Discussion

The current study investigated the role of SMA as a supra-modal hub in the sense of agency experience. We tested 25 healthy participants with an fMRI temporal judgment task [5] using stimuli of a different sensory modality compared to those used in our previous study [5]. Specifically, while in our previous experiment we employed visual action-feedback, the sensory effect produced by the voluntary action in this task was an auditory tone. Through this paradigm, we measured the intentional binding phenomenon, an implicit index of the sense of agency [6]. We showed different time-windows for the agency experience for the different sensory modalities. While in our previous study we reported an intentional binding effect only in the condition of a stringent 200 ms temporal contiguity between the action and the visual outcome, here we observed a sizeable time compression when the auditory outcome followed the action by 400 ms. At a neurofunctional level, we test the hypothesis that the activity of SMA covaried with the individually measured intentional binding effect in specific time-windows between actions and outcomes. Our results showed that the magnitude of the intentional binding effect was mirrored by meaningful brain activity in the anterior portion of SMA (pre-SMA). Importantly, the relationship between pre-SMA activity and intentional binding was significant only at the action-outcome time-window when there was a sizeable difference in the perceived time compression between the active and passive conditions for the specific sensory modality (400 ms for the auditory modality; 200 m for the visual modality).

These results provide novel insights concerning the link between the pre-SMA and the sense of agency experience whilst supporting previous studies’ conclusions. In particular, as we argue below, these findings confirm the sense of agency as a constructive phenomenon anchored to the motor system’s functioning and provide hints about the supra-modal relationship between pre-SMA and the sense of agency generation.

As mentioned, our behavioral results showed different time-windows for the arising of agency experience in the visual and auditory tasks. In particular, the agency experience seems to be “faster” in the visual than the auditory modality. While in our previous study we reported an intentional binding effect at 200 ms of delay between the action and the visual outcome, we observed here a later binding effect at 400 ms of delay between the action and the auditory outcome.

At first sight, this effect may be surprising since auditory processing is generally considered to be faster than visual processing [30]. Thus, one might expect participants to show an earlier intentional binding effect in the auditory than the visual task. However, the intentional binding effect does not represent a pure perceptual phenomenon. Rather, several sources of information, such as high-level causal beliefs and expectations, all modulate the binding (for a review see [31]). Therefore, while it looks implausible that there are precise time windows for the sense of agency in different sensory modalities, we speculate that specific expectations about the outcome may determine the particular time window for the sense of agency to occur. These outcome expectations could derive from our prior experiences with specific action-outcome associations and precisely from the repetition of the very same association until a pattern of regularity can be extracted. This is in line with previous studies showing that agency’s explicit and implicit measures are sensitive to the action-effect patterns to which people are exposed. For example, Haering et al. [32] showed that when participants were adapted to immediate action-effects, they felt less in control the longer was the delay between the action and the effect. In contrast, participants who were adapted to delayed effects showed the reversed result pattern and sensed less agency the shorter the delay between action and effect. Similarly, Kilteni et al. [33] showed that, after exposure to systematic delays, participants experienced less sensory attenuation (i.e., an implicit measure of agency whereby self-generated stimuli are perceived as less intense than externally-generated sensations [34]) for non-delayed self-generated touch, while they perceived as less intense and thus attenuated the delayed self-generated stimuli to which they were exposed. Therefore, the arising of a sense of agency seems to be tuned to time intervals that mimic the previous experience for a given action and its usual effects [5]. Crucially, one should consider that the visual stimuli used in our previous study had a clear link to real-life situations: indeed, a latency of about 200 ms is the one that can be measured in real life between the time when we press an electricity light-switch and the time that a conventional lightbulb takes to be fully on [35]. A binding effect at 200 ms of action-outcome delay is thus in line with the suggestion that a sense of agency emerges for action consequences that happen at action-outcome delays that are compatible with the expectations we made based upon our previous experiences [1,2,3]. Crucially, the auditory stimuli we adopted here (a pure tone) lacked any ecological meaning. As a consequence, participants could not rely on precise previous experiences with the stimulus, but only on a general expectation towards auditory outcomes. Importantly, while the same effect was described by Kuhn et al. [12], who showed a significant auditory intentional binding effect at 400 ms delay, other intentional binding experiments described the effect also at 250 ms action-tone delay [8]. It follows that we cannot disentangle here whether the results are due to different previous expectations or a more general difference in the sensory modality, with precise sense of agency time windows for each sensory modality. Our interpretation of the results thus remains speculative, even if more parsimonious. Future studies should orthogonalize those factors to explore the specific contribution (if any) of both the outcome sensory modality and the previous expectations towards it. Moreover, future studies should explore how those and other different sources of information are integrated to produce a coherent sense of agency, in accordance with the hypothesis that the sense of agency depends on a time window, starting from the onset of action and extending in time, within which different internal and external cues have to be integrated [36].

At a neurofunctional level, we tested the hypothesis that some brain regions’ activity covaried with the individually measured intentional binding effect in specific time-windows between actions and outcomes. We found that the activity of SMA, in its anterior part (pre-SMA), is linearly associated with the magnitude of the intentional binding effect. Remarkably, while in the previous study this association was significant at 200 ms of delay between the action and its visual effect, in this study, this association was significant at 400 ms of delay, i.e., when there was a sizeable difference in the perceived time compression between the active and passive conditions at the behavior level. No significant effects have been found at 200 and 600 ms.

This evidence provides validation of our approach, suggesting that the link between the pre-SMA activity and the agency experience is not trivial. Indeed, it rules out the possibility that our previous results were linked to the specific time course of the movement-related pre-SMA activation or they were somehow dependent on specific features of the stimuli we used to describe it. Conversely, the pre-SMA activity seems to be specifically anchored to the subjective feeling of agency towards self-generated outcomes across modalities, varying in time depending on the task’s specific features. This evidence provides support to the view that the brain mechanisms that give rise to our sense of agency are strictly motoric [1,2,3], completing the circle of a conceptual validation of the sense of agency as a phenomenon anchored to the functioning of the motor system [5,16]. Furthermore, it allows us to generalize the relationship between pre-SMA and the sense of agency as supra-modal. Previous studies have suggested a general link between the pre-SMA and the sense of agency experience by using visual [5], auditory [8,12], and tactile stimuli [10], raising the possibility of a supra-modal nature of this relationship. However, this research has mainly focused on the agency experience in a single sensory modality, and none of the aforementioned neuroimaging studies directly tested the role of the pre-SMA on the agency experience across different modalities through the same task and involving the same experimental participants. We showed here that the linear relationship between the pre-SMA activity and the magnitude of the intentional binding effect remains valid in spite of the sensory modality of the outcome. Even more importantly, it is tuned to the specific time-window in which the agency experience can be observed at a behavioral level for the specific task, suggesting a meaningful association between the pre-SMA activity and the experience of agency. However, it is worth noting that the pre-SMA activation is not exactly the same for the visual and auditory tasks. We observed a mesial to left hemispheric activation for the visual outcome and more mesial hemispheric activation for the auditory outcome. Therefore, we cannot describe the association pre-SMA-agency as anchored to a limited set of specific voxels. Rather, we can more in general suggest that the pre-SMA might be seen as a supra-modal *hub* in the agency generation. At the same time, the pre-SMA does not represent the unique brain area responsible for the agency attribution. Other brain areas were shown to be involved in this process [5]. Here we concentrate our analysis specifically on pre-SMA/SMA since in our previous study this area turned out to be causally involved in the sense of agency generation, while other brain regions did not show a similar causal relationship with the agency experience when TMS was applied during the task (see, for example, the parietal cortex) [5]. However, we cannot exclude that other brain regions might provide a similar supra-modal contribution to the sense of agency generation. Moreover, our study presents some other inevitable caveats. For example, the fact that in the passive condition, participants’ finger was pressed down by the experimenter. As a consequence, the active and the passive conditions are not completely comparable as there is additional tactile input in the passive condition. Other studies (see, for example, [37]) have managed to solve this problem by introducing elegant experimental manipulations in which the button mechanically went down. However, we still believe that those considerations were not such to prevent the observation of a meaningful relationship between fMRI activity in pre-SMA and the sense of agency measures for both visual and auditory action outcomes. 

## 5. Conclusions

Our results suggest that attributing consequences of self-generated movements to our actions is based on similar predictive mechanisms across different sensory modalities and that those mechanisms are strongly related to the functioning of the motor system.

## Figures and Tables

**Figure 1 brainsci-10-00825-f001:**
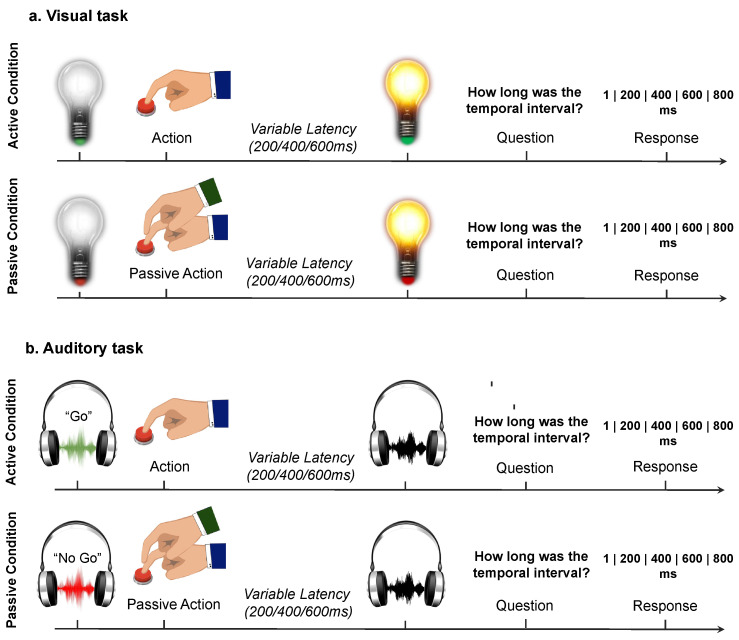
Experimental task. Graphical illustration of an experimental trial (for both active and passive conditions) for the visual (**a**) and the auditory (**b**) tasks. During the active trials, participants pressed a button with their right index finger at their own time after the presentation of the cue. In the passive trials, participants were instructed to stay still while an experimenter pressed their finger to produce a passive movement. In both conditions, the button press caused an action-consequence: the lightening of the lightbulb in the visual task, a pure tone in the auditory task. The consequence was presented after a variable delay of 200, 400 or 600 ms. Participants then judged the perceived time interval between their button press and the action-consequence (the lightening of the lightbulb or the tone).

**Figure 2 brainsci-10-00825-f002:**
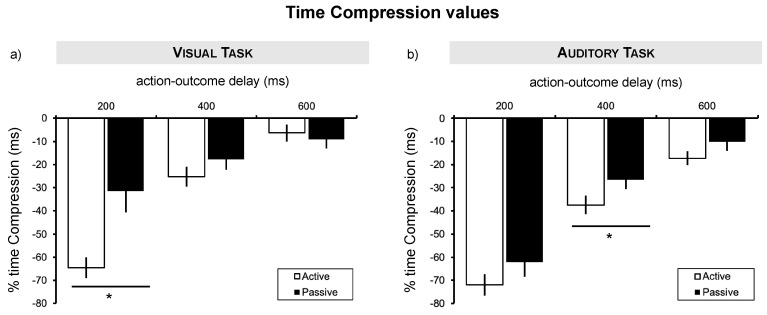
Behavioral results. (**a**) Behavioral results showing the intentional binding effect (greater time compression in the active than passive conditions) at 200ms action-outcome delay for the visual task. (**b**) Behavioral results showing the intentional binding effect (greater time compression in the active than passive conditions) at 400ms action-outcome delay for the auditory task. Error bars = standard error; asterisks indicate significant results at *p* < 0.05 Bonferroni corrected.

**Figure 3 brainsci-10-00825-f003:**
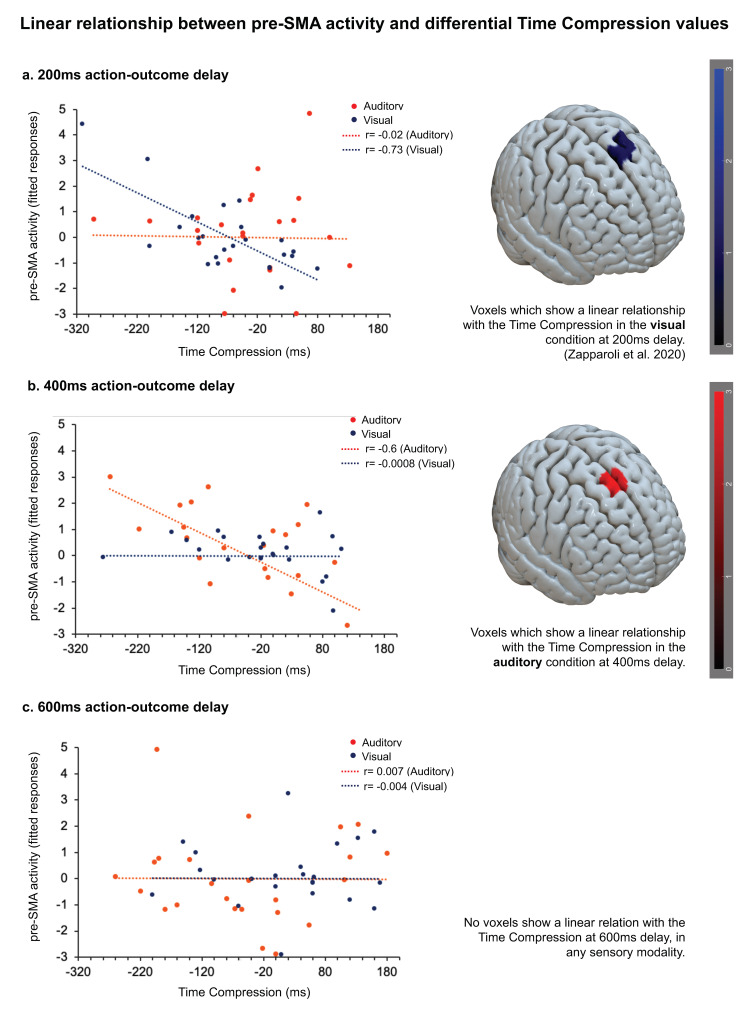
fMRI results. Linear relationship analysis between the pre-SMA activity during the task (fitted responses, active > passive conditions) and the differential TC values (active trials—passive trials) when the action-outcome delay was 200 ms (**a**), 400 ms (**b**) and 600 ms (**c**).

**Table 1 brainsci-10-00825-t001:** Planned post-hoc comparisons. Comparisons between time compression values in the active and passive conditions at different action-outcome delays, separately for each modality. For each comparison, we reported the mean difference, the standard error, the value of the statistic, the corresponding degrees of freedom and the associated Bonferroni-corrected *p* value. Asterisks indicate significant results at *p* < 0.05 Bonferroni corrected.

Comparisons											
Condition	Delay	Modality		Condition	Delay	Modality	Difference	Standard Error	Test	Degrees of Freedom	Bonferroni-Corrected *p*
Passive	200	Visual	-	Active	200	Visual	58.99	16.5	3.58	2614	0.002 *
Passive	400	Visual	-	Active	400	Visual	32.37	16.4	1.98	2614	0.3
Passive	600	Visual	-	Active	600	Visual	−16.35	16.2	1.01	2614	*p* > 0.99
Passive	200	Auditory	-	Active	200	Auditory	23.27	16.7	1.39	2614	*p* > 0.99
Passive	400	Auditory	-	Active	400	Auditory	42.64	16.3	2.62	2614	0.04 *
Passive	600	Auditory	-	Active	600	Auditory	32.67	16.4	2.00	2614	0.3

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
