# Peer review of "Predicting the Sensory Consequences of Self-Generated Actions: Pre-Supplementary Motor Area as Supra-Modal Hub in the Sense of Agency Experience"

_brainsci, 2020, doi:10.3390/brainsci10110825_

Round 1

Reviewer 1 Report

In this manuscript, Seghezzi and Zapparoli describe the results of an fMRI study designed to test the neural signature for a sense of agency in the SMA. Subjects had trials with a passive and active button press leading to delayed feedback and then they had to guess the length of the delay. The manuscript describes two sensory stimuli, visual and auditory, where the visual results were already published elsewhere and the suggested added value of this manuscript is in the replication of the finding in another sensory domain. The experimental protocol and the results of this study raise substantial concerns on whether the results are valid.

First, while the authors write that “fMRI data collected during the visual task are published in our recent study [5] and are inserted in the present paper for visualization only”, throughout the paper the results of the visual task are presented again, not as a reference but as an integral part of the results. Even more strikingly, while those should be the same results as in reference 5, the figures show huge differences. For example, the time compressions for the visual task in Fig2 are twice as much as those reported in the same plot from the same data in [5]. The TC preSMA correlation figure also looks different and even the cluster of voxels is different.

Second, the time compression in each trial is in jumps of 200ms (…,-200ms, 0ms, 200ms,…). However, it seems that for the fMRI analysis the authors model each subject as a single data point (averaging over the TCs?) as if the TC per subject is fixed. But there is clearly within-subject variability –  which lead to the spread of the reported TCs – but is completely ignored here. Since there are only 10 trials, one outlier trial can drive the subject’s performance. And since the entire story is based on a single correlation value for which no statistics was presented and seems to be driven by an outlier subject, the entire analysis may be driven by a single outlier trial.

Third, the main aim of the paper is to replicate the visual results in another sensory modality (auditory) to suggest that The SMA represents a supra-modal hub for agency. The results (if valid) suggest a fundamental difference between the 2 modalities in the time domain. Namely, the visual task shows the correlation between SMA activity and TC only for the 200ms delay while the auditory task shows it only for 400ms delay. The manuscript is not giving any explanation to this difference. While there are known feedback delay differences between sensory modalities which could potentially explain it, those are not addressed here and I’m not sure that they can explain 200ms difference. Moreover, to claim that the SMA represents such a supra-modal hub, one should be able to show that this correlation is unique for the SMA. However, the authors previous paper on this data set suggest that this relationship is evident across the brain (in the Superior front gyrus, Insula, Postcentral gyrus, Inferior parietal lobule, Precuneus, Hippocampus, Cerebellum, and more).

Minors

How long was the action-consequence (tone/lightbulb) presentation? What was the delay between it and the question?

What was the voxel size?

The results section is a long list of stats. It would read much better those would be in a table while the results text will tell the story from those numbers.

The authors should discuss the proprioceptive feedback form being touched and its potential effect on TC.

Reviewer 2 Report

This study investigates the role of the SMA in the sense of agency experience for visual and auditory action outcomes. They apply the intentional binding task in the scanner and combine the behavioral effect with the brain activity in the SMA. The results show an intentional binding effect at a 200ms delay for visual action outcomes and at 400ms delay for auditory action outcomes. For these delays there was a significant relationship between the intentional binding effect and the anterior part of the SMA (pre-SMA). The authors conclude that the pre-SMA represents a supra-modal hub in the sense of agency experience.

This manuscript is nicely written, well-structured and easy to follow. The research question is clearly motivated and the experimental design straight-forward. My main concerns belong to the incomplete methods and the interpretation of the results. Moreover, the figures could be improved to avoid any misunderstanding.

In the methods, there are several points unclear to me or completely missing. First, I find it rather unusual that the educational level is documented and so many tests for cognitive deficits are conducted. Was there any rational behind? This is a rather easy sensorimotor task, without requiring above average cognitive functioning. Second, it needs to be reported how the visual and auditory outcome conditions were recorded. Were the conditions blocked or interleaved? Were they presented in separate sessions on separate days or in one session on one day? Were they presented in counterbalanced order? Third, the fMRI analysis is only described for the auditory outcome condition. However, it remains unclear if the data (a) are pooled across the visual and auditory outcome or (b) if regressors of both the visual and the auditory outcome or (c) if only regressors for the auditory outcome were entered in the analyses (e.g. as regressors of no interest)? How were the fMRI data of the visual and auditory outcome conditions combined? More information is needed. Fourth, I don’t understand why the authors chose a between-subjects analysis and threated the conditions as between-subject factor. As all subjects performed both outcome conditions, this is not justified. I’m actually wondering if the results would look different if they would treat the conditions as within-subject factor, and thus consider the individual variability. Or maybe I misunderstand the analyses? Fifth, I miss information about the exact definition of the pre-SMA. Moreover, it is unclear if the anatomical SMA mask was applied on an individual or group brain level.

The claim that the pre-SMA is a supramodal hub can’t be derived from the present results. Here, I miss at least a conjunction analysis showing a significant overlap between the brain activity associated with the visual and the auditory action outcomes. In Figure 3, the pre-SMA activations look rather distinct with left hemispheric activation for the visual outcome and more medial to right hemispheric activation for the auditory outcome. The authors should provide the requested evidence or turn down their conclusions.

I found the interpretation of the different delay effects for the visual and auditory outcome conditions very speculative. The authors claim that a rapid action effect is learned from everyday experience for the visual outcome and that this visual action effect (light follows a button press) is more natural than the auditory action effect (tone follows a button press). This unnatural action effect should then lead to longer information processing and therefore for a later binding effect. I don’t fully agree with this argumentation. If I type on my keyboard I also here a tone immediately after I pressed the key. Or if I ring a door bell I will here very different (rather arbitrary tones) as the outcome of my button press. I have to admit that I don’t have a good explanation for the differences in delay. One could argue that the different delays presented in the task actually impede to establish a binding effect; and especially in the auditory domain because it is temporally more precise than the visual one. Anyway, this is an interesting finding that should be reported. Based on the present study a clear explanation can’t be given. I recommend that the authors shortened this part and just briefly mention their idea about ecological meaning. In addition, they may come up with alternative explanations that could be tested in future studies. Also a discussion of learned temporal action-outcome relationships would add to it (e.g. Kilteni et al., 2019, ELife, 8, e42888). In addition, the paper from Farrer et al., 2013 (Consciousness and Cognition, 22(4), 1431-1441) could be integrated.

Minor points

Methods: In the passive condition, the finger of the participant was pressed down by the experimenter. This manipulation is not as elegant as in other studies in which the button mechanically went down (e.g. van Kemenade et al., 2016). Therefore, the active and the passive conditions are not completely comparable as there is additional tactile input in the passive condition. This limitation needs to be mentioned.

Methods: Please add for how long the action effects (light and tone) were presented. This information is missing.

Behavioral results: Please explain what the significant results mean in the first paragraph of this section. Please note that post-hoc t-tests are not required if the factor has only two levels. The results between line 256 and 270 are really hard to read because the brackets with the statistical details are so long. I recommend to expand Table 1 and add these values to the table. Some are already reported there.

fMRI results: It’s unclear to me why the results are ordered in this way - 400ms, 200ms, and 600ms – and not like in a logical way as in the figure 3 – 200ms, 400ms, and 600ms. There are also errors in referring to the figure, 3.2.1 should refer to 3b and not 3a, and 3.2.2 should refer to 3a and not 3b. Lines 304-305: Please add in the brackets “200ms vs. 400ms:” and “200ms vs. 600ms:” to make clear that the results belong to the delay comparisons.

Figure 1: From the figure it looks like participants pressed the response keys with their index finger. In the text it is stated that the response was actually given with all five fingers. Please adjust the figure accordingly.

Figures 2 and 3: Please check the fonts. There seems to be some formatting or conversion error (e.g. headings “Action-outcome delay (ms)” in Fig. 2)

Figure 3: It took me some time to understand that the plot in 3a belongs to the results of the published study. An additional heading above 3a “Visual outcome (see Zapparoli et al., 2020)” and above 3b “Auditory outcome” would be helpful. Moreover, the figure capture could be shorten: “… , (b) 400ms of action-outcome delay, (c) 600ms of action outcome delay.” The blue and red dots are explained in the legend and doesn’t need to be explained again in the caption.

Lines 74-75: It reads like the authors performed two experiments, one behavioral experiment (outside the scanner) and one fMRI experiment. Please adjust.

Line 216: delete s in others

Line 218: that -> than

Line 316: In the methods, 24 participants were part of the final sample - not 25?

Round 2

Reviewer 2 Report

The authors sufficiently answered my questions and added all missing details to the manuscript. I appreciate the clarification of the methods and improvement of the figures. Also the discussion is now more adequate given the results.

In the corrected sections, there are some minor points that should be corrected.

line 224: delete "and"

lines 308-309: The authors report that the coefficient for the 400ms of action-outcome delay was significantly higher than the same coefficients calculated for the 200 and 600ms of action-outcome delay. So I would expect two comparisons and therefore two z- and p-values: (1) comparison of 400ms vs. 200ms delay and (2) comparison of 400ms vs. 600ms delay. The one z- and p-value is confusing. Please clarify.

line 435: Please delete "but unfeasible in an fMRI setting". There is actually a paper from the same group (citation [37]) under review in which they used a passive button device in the scanner. They applied air pressure to automatically move the button down.
